# COVID-19 in Low- and Middle-Income Countries (LMICs): A Narrative Review from Prevention to Vaccination Strategy

**DOI:** 10.3390/vaccines9121477

**Published:** 2021-12-14

**Authors:** Sansone Pasquale, Giaccari Luca Gregorio, Aurilio Caterina, Coppolino Francesco, Passavanti Maria Beatrice, Pota Vincenzo, Pace Maria Caterina

**Affiliations:** Department of Woman, Child and General and Specialized Surgery, University of Campania “Luigi Vanvitelli”, 80138 Napoli, Italy; lucagregorio.giaccari@gmail.com (G.L.G.); Caterina.aurilio@unicampania.it (A.C.); francesco.coppolino@unicampania.it (C.F.); MariaBeatrice.PASSAVANTI@unicampania.it (P.M.B.); vincenzo.pota@inwind.it (P.V.); mariacaterina.pace@unicampania.it (P.M.C.)

**Keywords:** COVID-19, low-and middle-income countries (LMICs), infection prevention, triage, vaccine

## Abstract

The management of the COVID-19 pandemic represents a challenging process, especially for low- and middle-income countries (LMICs) due to the serious economic and health resource problems it generates. In this article, we assess COVID-19 situation in LMICs and outline emerging problems and possible solutions. The prevention and control of COVID-19 would be based on focused tests exploiting those systems (e.g., GeneXpert^®^) already used in other scenarios. This would be less stressful for the healthcare system in LMICs. Avoiding close contact with people suffering from acute respiratory infections, frequent handwashing, and avoiding unprotected contact with farm or wild animals are recommended infection control interventions. The appropriate use of personal protective equipment (PPE) is required, despite its procurement being especially difficult in LMICs. Patients’ triage should be based on a simple and rapid logarithm to decide who requires isolation and targeted testing for SARS-CoV-2. Being able to estimate which patients will develop severe disease would allow hospitals to better utilize the already limited resources more effectively. In LMICs, laboratories are often in the capital cities; therefore, early diagnosis and isolation become difficult. The number of ICU beds is often insufficient, and the equipment is often old and poorly serviced. LMICs will need access to COVID-19 treatments at minimal prices to ensure that all who need them can be treated. Year-to-date, different vaccines have been approved and are currently available. The main obstacle to accessing them is the limited ability of LMICs to purchase significant quantities of the vaccine.

## 1. Introduction

**Background**. On 31 December 2019, the Chinese government reported several cases of pneumonia of unknown etiology detected in Wuhan, the capital city of Hubei province. A week later, a new type of coronavirus (SARS-CoV-2) was identified as the etiological cause of this severe acute respiratory syndrome. On 11 March 2020, the World Health Organization (WHO) declared COVID-19 a pandemic. For the WHO, a pandemic is the spread of a new disease globally [1]. According to the Centers for Disease Control and Prevention (CDC), pandemics occur when “*new viruses emerge which are able to infect people easily and spread from person to person in an efficient and sustained way*” [2].

At the time this article was written, there had been 243,857,028 confirmed cases of COVID-19, including 4,953,246 deaths [3]. According to the WHO, the distribution of confirmed cases among different regions is as follows: Eastern Mediterranean 16,267,384, Europe 75,402,101, Americas 92,930,797, South-East Asia 43,830,719, Africa 6,136,680, and Western Pacific 9,288,583 (see Figure 1) [3].

Different countries vary widely in terms of their capability to prevent, detect, and respond to outbreaks of an infectious disease [4].

The World Bank has historically classified countries according to their *per capita gross national income* (GNI) into three groups: *high-income countries* (or HICs), *middle-income countries* (or MICs), and *low-income countries* (or LICs) [5,6,7,8,9]. In 2020, LICs were 32, suffering high rates of illnesses and infections due to the lack of clean water, low sanitation levels, malnutrition, and the lack of access to quality medical care. Approximately 5 billion people lived in MICs, representing over 70% of the world population. There are a total of 105 MICs. Only 77 countries were classified by the World Bank as HICs.

In the COVID-19 era, the problems all countries, but especially those with serious economic and health resource challenges, had to face became evident.

**Aims.** At this writing, no curative treatment has been found for COVID-19, and the vaccination campaign is in progress. To our knowledge, the literature on diverse aspects of the disease and therapeutic implications in low- and middle-income countries (LMICs) is limited. This narrative review aims to summarize major research elucidating the COVID-19 situation in LMICs to outline emerging problems and possible solutions focused on prevention and vaccination strategies.

## 2. Materials and Methods

We conducted a literature search to identify publications reporting on COVID in LMICs and we defined inclusion criteria as follows:publications reporting on COVID19 prevention and control, triage strategies, available infrastructure including Intensive Care Unit (ICU), treatment and vaccines;publications written in English.

The search combination “*low- and middle-income countries (LMICs)*”, “*developing countries*”, and “*COVID-19*” was entered in the main electronic databases (EMBASE, PubMed, Google Scholar, and The Cochrane Library—CENTRAL). Other relevant studies were identified from the reference lists and the online sites of national and international agencies or societies (CDC, www.cdc.gov (accessed on 15 October 2021); NIH, www.nih.gov (accessed on 15 October 2021); WHO, www.who.int (accessed on 15 October 2021)). The search was performed in October 2021, and we included all publications up to the end of September 2021. Two researchers (P.S. and L.G.G.) screened the titles and abstracts to identify the keywords. The selected papers were read in full by the two independent reviewers, and a third reviewer (M.C.P.) reviewed the decisions, verifying the inclusion criteria.

## 3. Results

A total of 38 full texts were eligible for inclusion. As shown in Figure 2, the flow diagram reports the results from the literature search and the study selection process.

From a total of 2852 records screened, only 38 articles were included in this review. In Table 1 and Table 2, all studies are presented in the order found in the main text, with a brief description.

### 3.1. Infection Prevention and Control

Fundamental interventions to control the COVID-19 infection are: (1) a rapid identification of the infected individuals; (2) an effective response to patients and the community; (3) preventive interventions [6].

The WHO has emphasized the importance of diagnostic tests in tracking and managing COVID-19. Detection is currently based on real-time reverse transcriptase-polymerase chain reaction (RT-PCR) in nasopharyngeal samples [7].

The United States has conducted more COVID-19 tests than any other country. On 7 March 2021 the COVID-19 tests performed in the U.S.A. were 363,825,123 [8]. This level of testing starkly contrasts with the one available in LMICs [9].

In LMICs, a focused testing on selected patients instead of a random screening would be less stressful for the healthcare system [10]. COVID-19 tests should target patients with coexisting diseases or atypical presentations, pregnant women, and health workers. For other patients, a clinical case definition based on symptoms and radiology results should be adopted, such as that proposed by the Haitian Ministry of Public Health and Population [11].

Other problems are the lack of a country-based testing plan, the lack of sufficiently trained staff for performing RT-PCR, and insufficient supplies of the reagents and kits for nucleic acid extraction and molecular detection, needed to perform a high number of tests for SARS-CoV-2 [7]. Rapid test kits are an option to allow LMICs to perform diagnostic tests faster [10].

In 2009, the World Health Organization Regional Office for Africa (WHO AFRO) launched the Stepwise Laboratory Quality Improvement Process Towards Accreditation (SLIPTA) program with the aim to strengthen laboratories’ compliance with international standards through training and mentoring [12]. Nevertheless, the mode of testing for COVID-19 in Africa is through reference laboratories and central laboratory testing. Sub-Saharan Africa (SSA) countries can take advantage of GeneXpert^®^, a multi-disease diagnostic platform used initially to test tuberculosis (TB) and later adapted for human immunodeficiency virus (HIV) and Ebola [13]. The GeneXpert^®^ is a molecular testing platform which can be located in all laboratories for immediate diagnostics [14]. It processes samples onsite, reducing transportation time and test waiting times to only 45 min. It has been successfully used in Madagascar, showing that the use of the GeneXpert^®^ platform to screen patients with SARS-CoV-2 in LMICs is relevant and achievable and should be adopted in countries with difficult access to laboratories and an already pre-existing GeneXpert^®^ network [15]. In order to slow down the spread of COVID-19, the WHO recommends infection control interventions to reduce the risk of transmission, in particular, avoiding close contact with people suffering from acute respiratory infections, frequent hand washing especially after direct contact with infected people or their environment, and avoiding unprotected contact with farm or wild animals [16]. Worldwide, governments have established regulations that require social distancing, the closure of non-essential businesses, travel restrictions and, in many cases, quarantine. Although these measures are necessary for public health, social restrictions are difficult to realize in LMICs due to money-related livelihood problems [17]. Furthermore, a complete commercial shutdown like those imposed in China, Europe, or the United States is not feasible when a day without work is tantamount to a day without food [11].

Appropriate personal protective equipment (PPE) is required by all available guidelines for the management of COVID-19 patients. PPE is in enormous demand around the world, and its procurement will thus prove especially difficult in LMICs [18]. Some low-cost suggestions were proposed for creating or extending PPE. In emergency situations, raincoats and windcheaters were used as gowns, while swimming caps, goggles, and transparent paper were used as PPE [19].

### 3.2. Triage

According to the WHO, an efficient triage of patients with COVID-19 will help the national planning and case management system response to manage the patients influx, addressing the medical resources needed to efficiently support critical patients and protect the safety of healthcare workers [20].

Several guidelines for triage were developed in various countries, with the aim of reducing the burden on those who must determine which patient has access to the limited resources [21]. In LMICs, ethical challenges are emerging due to pre-existing problems and the shortage of medical resources. Triage criteria are based on community values, reflecting society’s moral standards and ideals.

In resource-limited settings, it is necessary to adopt a simple and rapid logarithm to decide who requires isolation and targeted testing for SARS-CoV-2. The most recent recommendations suggest that initial screening to identify individuals with COVID-19 include signs and symptoms based on standard case definitions of COVID-19 disease [22]. In these countries, other endemic febrile illnesses are particularly common, and their presentation could easily be similar to that of COVID-19. Clinician should consider these diseases as part of the comprehensive clinical diagnostic process.

Being able to estimate which patients will develop severe disease would allow hospitals to better utilize already limited resources more effectively. In this way, hospitals could stop admitting patients at low risk of deterioration by avoiding unnecessary treatment. A severity score based on the COVID-19 disease definition should be performed for all suspected or confirmed COVID-19 cases before their access to the emergency department [22].

The approach should use the available technology. In LMICs, there are several problems regarding the availability of radiologic diagnostic modalities, the status of the machines, and the availability of clinical staff who are skilled in performing and interpreting the exams [22].

In Uganda, Ayebare et al. proposed an algorithm triage based on common COVID-19 symptoms, such as fever or cough. When these symptoms are found in combination with the epidemiological risk, patients are isolated by adopting appropriate measures for infection prevention and control, and the SARS-CoV-2 test is performed [23].

In India, Nayan et al. proposed triaging by medical history and symptom-based test probability assessment for COVID-19. Only eligible patients undergo further investigations. Due to the poor healthcare resources, this approach is fast, saves important resources, and decreases the risk of transmission of the infection to the health workers [24].

### 3.3. Infrastructure and Intensive Care Unit

The COVID-19 pandemic represents a major challenge for healthcare services worldwide. The surge of patients with COVID-19 is putting unprecedented stress on existing services, infrastructure, and healthcare workers in developing countries.

In LMICs, laboratories are often located in the capital cities, so that early diagnosis and isolation becomes difficult [25]. Furthermore, the infrastructures for the screening and treatment of COVID-19 are not separate from the ones devoted to non-COVID-19 healthcare, facilitating the spread of the infection.

The number of hospital beds and health workers is generally lower compared to that in HICs [26]. The WHO reports only 0.8 hospital beds per 1000 people in LICs and 2.3 in MICs. On the other hand, HICs have 5.3 hospital beds per 1000 people [27]. According to the WHO, 90% of LICs have fewer than 10 medical doctors per 10,000 people, compared to only 5% of HICs. Up to 93% of LICs have fewer than 40 nursing personnel per 10,000 people, compared to only 19% of HICs [27].

Whenever published data are available, the number of ICU beds is insufficient with respect to the population of LMICs [28]. The most recent data available from the WHO indicate that Africa has fewer than 5000 ICU beds, corresponding to five beds per one million people. In Europe, by comparison, there are 4000 beds per one million people.

In other countries, heterogeneous regional distribution and payer-based access are major barriers for the equitable delivery of critical care, despite a sufficient number of ICU beds [29]. In Brazil, there are about 100 ICU beds for every million people, but their distribution is not uniform among the different regions: the states of São Paulo (~18,000 ICUs), Rio de Janeiro (~7000 ICUs), and Minas Gerais (~6000 ICUs) concentrate the ICUs, and most of them are located in the capital cities [30]. Furthermore, long distances and high transportation costs commonly result in delayed presentation of critically ill patients.

The ICU equipment is often old and poorly serviced. Mechanical ventilators tend to be old, and many hospitals do not have oxygen or medical gas to drive them [31,32]. Generally, equipment maintenance is poorly performed if available, and funding for capital development is limited. When funding is available, the procurement system is plagued by corruption, leading to a fraudulent assignment.

WHO developed a document providing recommendations, technical guidance, standards, and minimum requirements for setting up and operating Severe Acute Respiratory Infection (SARI) treatment centers in LMICs. Important guidance in setting up a SARI treatment center in LMICs is given in a practical manual by the WHO entitled “*Severe acute respiratory infections treatment centre: practical manual to set up and manage a SARI treatment centre and a SARI screening facility in health care facilities*” [33].

### 3.4. Treatment

Unfortunately, there are no medications which were demonstrated to successfully treat COVID-19. Since SARS-CoV-2 replication is the cause of many of the clinical manifestations of COVID-19 [34], antiviral drugs are being evaluated to treat COVID-19. Antiviral therapies include the nucleotide analogue *remdesivir*, originally developed as a potential treatment for Ebola [35], the HIV protease inhibitor *lopinavir/ritonavir* [36], and the antimalarials *chloroquine* and *hydroxychloroquine* [37]. In addition, treatments for improving lung function and reducing inflammation, such as *corticosteroids* and *tocilizumab*, are being evaluated in clinical trials.

To date, remdesivir is the only antiviral recommended for use in hospitalized patients with COVID-19 [38]. It was first described in the literature in 2016 as a potential treatment for Ebola [35]. Remdesivir has been studied in several clinical trials for the treatment of COVID-19. The Food and Drug Administration (FDA) granted an Emergency Use Authorization (EUA) on 1 May 2020 [39]. Remdesivir was subsequently granted full approval as a COVID-19 treatment on 22 October 2020 [40].

The HIV antiretroviral combination lopinavir/ritonavir has also been proposed as a treatment for COVID-19 [41]. Although it has in vitro activity against SARS-CoV, higher than tolerable levels of the drug might be required to achieve inhibition in vivo [42,43]. Lopinavir/ritonavir and the other HIV protease inhibitors are not recommended for the treatment of COVID-19 according to the COVID-19 Treatment Guidelines Panel [38].

Chloroquine, an older drug used as an antimalarial, and its derivative hydroxychloroquine have become popular in an effort to find an effective treatment for COVID-19. In vitro, hydroxychloroquine shows superior activity against COVID-19 compared with chloroquine [44]. Despite the initial excitement about the potential significant effectiveness of chloroquine and hydroxychloroquine for treating COVID-19, recent studies suggest little to no benefits in using these drugs for COVID-19 treatment. Nevertheless, they are still likely to be considered as a potential therapy by many in the fight against COVID-19, especially in LMICs [45].

Severe forms of COVID-19 can lead to a systemic inflammatory response with sequelae of lung injury and multisystem organ dysfunction. Corticosteroids can be considered a therapeutic option for critically ill patients. Dexamethasone is a low-cost corticosteroid and it has recently been shown to significantly reduce mortality in patients most severely affected by COVID-19 [46].

LMICs will need access to these treatments at minimal prices to ensure that all who need them can be treated. A recent report shows that COVID-19 treatment drugs could be produced at very low prices, from $1 to $29 per course [47]. Some of these drugs are already offered as generics at prices close to the production cost, in LMICs. The Global Fund for AIDS, Tuberculosis, and Malaria (GFATM) and the President’s Emergency Plan for AIDS Relief (PEPFAR) are organizations whose purpose is to provide mass treatment for HIV, tuberculosis, and malaria at prices close to production costs [48,49]. In the current context, this system allows LMICs to access drugs to treat COVID-19 at affordable prices while mitigating the impact of the pandemic on other health programs.

Oxygen therapy and ventilation are two main non-pharmacological interventions used for COVID-19 patients.

In 2015, the “*Lancet Commission on Global Surgery*” revealed that approximately 24% of hospitals in LMICs lack oxygen supply [50].

Supplemental oxygen can be provided using simple nose prongs or face masks. Other modalities require specific equipment, which can be expensive and difficult to acquire, especially in LMICs. In the current state of the pandemic, several alternatives for respiratory support in the shortage of official devices are proposed, such as cheap CPAP helmets obtained by adapting diving mask [51].

In resource-limited settings, the provision of quality mechanical ventilation is challenging for several reasons [52]. First, a small number of the ICU beds are equipped with mechanical ventilators. Second, equipment maintenance can be problematic due to the frequent need to reuse single-use components, poor access to consumables such as heat and moisture exchangers and suction catheters, poor access to spare ventilator parts like flow meters, unreliable oxygen supply, and inconsistent electricity. Third, highly skilled capabilities are needed for quality ICU care in order to provide an adequate ventilatory support in severe cases of COVID-19. This will require extended staff training, not always possible. Finally, in emergency situations, a single ventilator is used to assisted multiple patients. This leads to an unequal delivery of gas volumes and pressures to the individual patients, compromising the individualized ventilator settings needed for optimal care.

In the context of the high importance of oxygen therapy for patients with severe COVID-19, the WHO provides a useful guidance on the different oxygen sources, entitled “*Oxygen sources and distribution for COVID-19 treatment centres*” [53].

### 3.5. Vaccine

The development of a safe and effective vaccine for COVID-19 has been a major global challenge. To date, there are 12 approved vaccines for emergency use or with full authorization; many more are in development. On 11 December 2020, the Food and Drug Administration (FDA) issued the first EUA for “*tozinameran*”, the COVID-19 vaccine produced by Pfizer-BioNTech. The U.S.A. has approved a second COVID-19 vaccine, the “*mRNA-1273*” manufactured by Moderna. The European Medicines Agency (EMA) approved the Pfizer-BioNTech COVID-19 vaccine on 21 December 2020. The EMA authorized Moderna’s vaccine distribution on 6 January 2021. “*AZD1222*”, developed by Oxford University and AstraZeneca, is the third COVID-19 vaccine that EMA has recommended for authorization. On 28 February 2021, the FDA recommended the use of another vaccine produced by Janssen Pharmaceuticals and Johnson & Johnson. EMA approved the COVID-19 Vaccine Janssen on 11 March 2021.

According to *People’s Vaccine Alliance*, in the context of pandemic COVID-19, the main obstacle to an adequate vaccination campaign is the ability of LMICs to procure significant quantities of the vaccine. As of 15 November 2020, several countries have pre-ordered a total of 7.48 billion doses of COVID-19 vaccines; 51% of these doses will go to HICs, which correspond to only 14% of the world’s population [54].

Currently, 52.2% of the world population has received at least one dose of a COVID-19 vaccine. As of 15 November 2021, a total of 7,307,892,664 vaccine doses have been administered [3], and 31.32 million doses are now administered each day. Only 4.6% of people in LMICs have received at least one dose. As shown in Figure 3, a large difference in the vaccine doses administered emerges between the various countries.

The COVID-19 Vaccine Global Access Facility (COVAX)—led by the Global Alliance for Vaccines and Immunizations (Gavi), the Coalition for Epidemic Preparedness Innovations (CEPI), and the WHO—attempts to facilitate the equitable access to and distribution of these vaccines to protect people in all countries [55,56].

The initial goal is to have 2 billion doses available by the end of 2021, with the aim of vaccinating up to 20% of the population by the end of the year.

## 4. Discussion

The impact of COVID-19 on LMICs is expected to be significantly worse compared to that on most developed countries with respect not only to the impact on the health crisis in the short-term but also to the long-term social and economic effects.

The coronavirus pandemic has overwhelmed health systems in Europe and North America. HICs have experienced shortages of health personnel, ventilators, personal protective equipment and testing capacity. As highlighted in our review, it is reasonable to expect a worse outcome in LMICs where medical resources are scarce.

The WHO has recommended physical distancing to control the spread of the virus, but in countries where people live in confined spaces and lack running water for personal hygiene, these measures are difficult to adopt. It is therefore crucial to take prompt measures to ensure public health, preventing and controlling the spread of COVID-19. In addition, guidelines for patients’ triage must be established to address the medical resources needed to efficiently support critical patients and protect the safety of healthcare workers. All these initiatives are essential, given the limited resources or their unequal distribution in LMICs.

An established therapy has not yet been found, and most of the drugs adopted in HICs are often inaccessible due to their high cost. It is important to encourage the access to high-quality drugs at affordable prices to reinforce the response to COVID-19 and mitigate the impact of the pandemic on others health programs. Multiple variants of the virus that causes COVID-19 are circulating globally. Particularly in LMICs, the virus has the potential to mutate indefinitely and survive due to the high percentage of the unvaccinated population. It is mandatory to aim for a vaccination campaign that can provide adequate coverage.

## 5. Conclusions

The COVID-19 pandemic can be limited when pandemic response strategies and tactics are implemented early. The role of HICs, international institutions (e.g., WHO), and various humanitarian organizations, already operating in LMICs, is crucial in this common fight against COVID-19. Further efforts are needed to increase vaccine availability for a large number of LMICs in the coming months.

## Figures and Tables

**Figure 1 vaccines-09-01477-f001:**
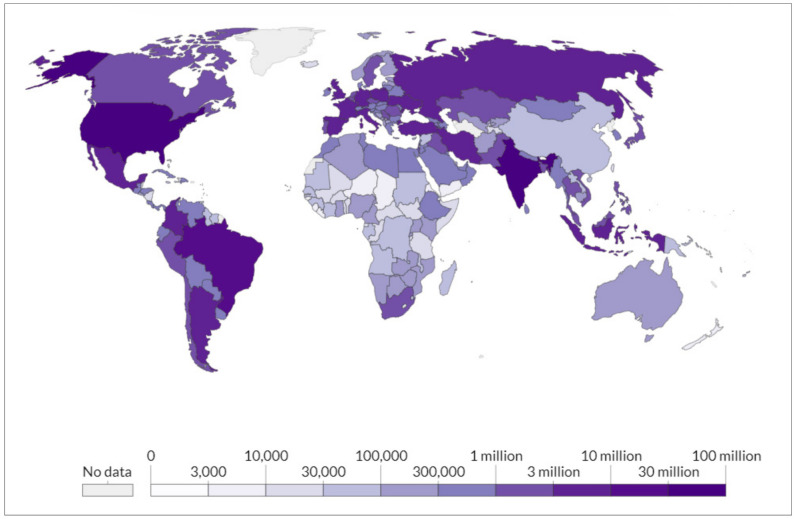
Total COVID-19 cases (2 December 2021). OurWorldInData.org. Available online: https://ourworldindata.org/coronavirus (accessed on 2 December 2021).

**Figure 2 vaccines-09-01477-f002:**
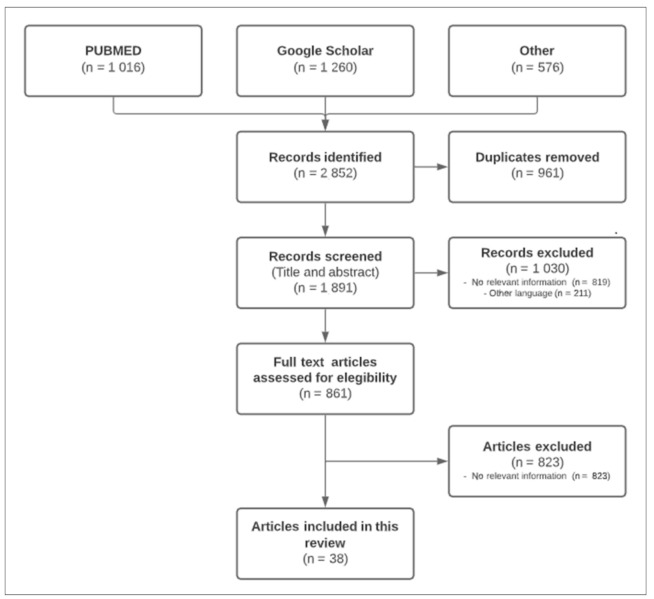
Flow diagram of the study selection process.

**Figure 3 vaccines-09-01477-f003:**
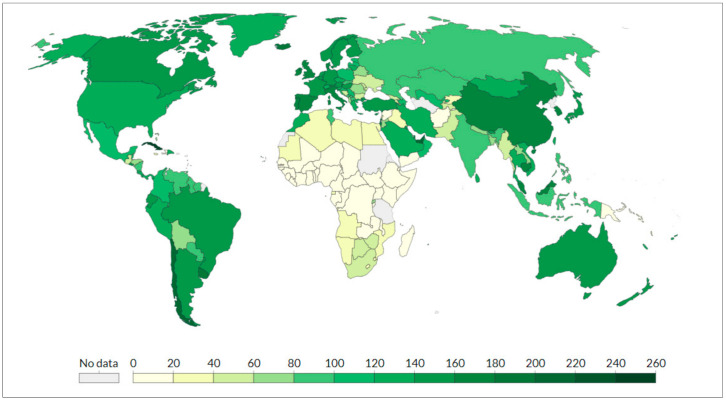
Total doses administered per 100 population worldwide (2 December 2021). OurWorldInData.org. Available online: https://ourworldindata.org/coronavirus (accessed on 2 December 2021).

**Table 1 vaccines-09-01477-t001:** Infection prevention and control; triage; infrastructure and Intensive Care Unit (ICU).

Author	Country/Region	Topics
Anjum F.R. et al. [6]	–	Interventions to control COVID-19 infection.
Tang Y.W. et al. [7]	–	Laboratory diagnosis of COVID-19 infection.
Monjur M.R. et al. [9]	Bangladesh	Lack of a prevention policy in Bangladesh.
Siow W.T. et al. [10]	LMICs	COVID-19 infection management in resource-limited settings.
Rouzier V. et al. [11]	Haiti	Advice for LMICs on managing COVID-19 infection.
Quaresima V. et al. [13]	Sub-Saharan Africa	Africa Task Force for Novel Coronavirus (AFCOR) activities for COVID-19 infection management.
Rakotosamimanana N. et al. [15]	LMICs	GeneXpert for the diagnosis of COVID-19 infection in LMICs.
Lai C.C. et al. [16]	–	Interventions to control COVID-19 infection.
Krishnakumar B. et al. [17]	India	Interventions to control COVID-19 infection in India.
Wang M.W. et al. [18]	–	Lack of protective mask.
Sudhir A. et al. [19]	India	Interventions to control COVID-19 infection in India.
WHO [20]	LMICs	Algorithm for COVID-19 triage and referral in LMICs.
Joebges S. et al. [21]	–	Ethics guidelines on COVID-19 triage.
Barros L.M. et al. [22]	LMICs	Recommendations for identification and triage of patients with COVID-19 in LMICs.
Ayebare R.R. et al. [23]	Uganda	Algorithm for COVID-19 triage and referral in Uganda.
Nayan N. et al. [24]	India	Algorithm for COVID-19 triage and referral in India.
Atreya A. et al. [25]	Nepal	Health resources in Nepal during COVID-19.
Harris C. et al. [26]	–	State of clinical research for COVID-19 and recommendations for the implementation of standardised protocols.
Murthy S. et al. [28]	LMICs	Intensive Care Unit capacity in LMICs during COVID-19.
Salluh J.I.F. et al. [29]	Brazil	Intensive Care Unit capacity in Brazil during COVID-19.
Marson F.A.L. et al. [30]	Brazil	COVID-19 in Brazil.
Federal Republic of Nigeria [31]	Nigeria	Medical oxygen in health facilities in Nigeria.
Replabic of Uganda [32]	Uganda	Medical oxygen implementation plan in Uganda.
WHO [33]	LMICs	Severe acute respiratory infections (SARI) treatment centre.

**Table 2 vaccines-09-01477-t002:** Treatment and vaccines.

Author	Topics
National Institutes of Health [38]	COVID-19 Treatment Guidelines.
Dagens A. et al. [41]	COVID-19 Treatment Guidelines.
Liu X. et al. [42]	Predicted commercial medicines as potential inhibitors against COVID-19.
Chen F. et al. [43]	Antiviral agents activity against COVID-19.
Yao X. et al. [44]	Hydroxychloroquine activity against COVID-19.
Saha Bket al. [45]	Antimalarials activity against COVID-19.
Siemieniuk R. et al. [46]	COVID-19 Treatment Guidelines.
Hill A. et al. [47]	Minimum costs to manufacture new treatments for COVID-19.
Bibiano-Guillen C. et al. [51]	Adapted Diving Mask (ADM) device as respiratory support with oxygen output during COVID-19
Dondorp A.M. et al. [52]	Respiratory support in LMICs during COVID-19.
WHO [53]	Oxygen sources and distribution for COVID-19 in LMICs.
So A.D. et al. [54]	COVID-19 vaccines for global access.
McAdams D. et al. [55]	COVID-19 vaccine Global Access Facility (COVAX).
Lie R.K. et al. [56]	COVID-19 vaccine allocation.

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
