# Peer review of "COVID-19 in Low- and Middle-Income Countries (LMICs): A Narrative Review from Prevention to Vaccination Strategy"

_vaccines, 2021, doi:10.3390/vaccines9121477_

Round 1

Reviewer 1 Report

The authors addressed the disparities of COVID vaccines in low and middle-income countries. I have some comments for the authors to improve the current manuscript.

  1. Include a flow chart showing the number of papers reviewed and the number of papers excluded and included by criteria.
  2. Although this is a review paper, authors collected COVID-related data and visualize in the map. Then, include another section for this to show the source of datasets, how those rates were calculated and visualized in the map in the methods section.
  3. Remove "5. Patents" on the page 9.

Author Response

#REVIEWER 1

  • Include a flow chart showing the number of papers reviewed and the number of papers excluded and included by criteria. → ADDED. We added also Table 1A,1B,1C, 1D and 1E to present all the studies included in our review with a brief description for each case.

  • Although this is a review paper, authors collected COVID-related data and visualize in the map. Then, include another section for this to show the source of datasets, how those rates were calculated and visualized in the map in the methods section. → The 2 maps are published online at OurWorldInData.org [https://ourworldindata.org/coronavirus], as reported in the text. We think they help the reader (expecially them not from a scientific field) to understand the COVID-19 problems in LMICs.

  • Remove "5. Patents" on the page 9. → REMOVED.

With regard,

the authors.

Reviewer 2 Report

The topic is of course important. I would suggest however some important changes. The introduction is very historical and seems to lack a clear problematic, hypotheses. What is the added value of the proposed paper in the  field?

The maps often look like maps found in newspapers or ourworldindata. They do not bring anything new.

By contrast what seems to me to be the most important added value the synthesis of the literature is presented in text, often a bit lengthy, but there is no table. A synthtic table distilling the main findings perhaps suggestions gaps... would be interesting.

Author Response

#REVIEWER 2

  • What is the added value of the proposed paper in the field? → MODIFIED: “At this writing, no curative treatment has been found for COVID-19 and the vaccination campaign is in progress. To our knowledge, there is few summarized literature of diverse aspects of the disease and therapeutic implications in low and middle-income countries (LMICs). This narrative review aims to summarize major research elucidating the COVID-19 situation in LMICs to outline emerging problems and possible solutions focusing on prevention and vaccination strategies. ”

  • The maps often look like maps found in newspapers or ourworldindata. They do not bring anything new. → MODIFIED. We have kept 2 maps on infections and vaccinations. The 2 maps are published online at OurWorldInData.org [https://ourworldindata.org/coronavirus], as reported in the text. We think they help the reader (expecially them not from a scientific field) to understand the COVID-19 problems in LMICs.

  • By contrast what seems to me to be the most important added value the synthesis of the literature is presented in text, often a bit lengthy, but there is no table. A synthtic table distilling the main findings perhaps suggestions gaps... would be interesting. → MODIFIED: We added also Table 1A,1B,1C, 1D and 1E to present all the studies included in our review with a brief description for each case.

With regard,

the authors.

Round 2

Reviewer 2 Report

the authors have answered my comment

english is problematic and should be improved